# Long Electrical Stability on Dual Acceptor p-Type ZnO:Ag,N Thin Films

**DOI:** 10.3390/mi15060800

**Published:** 2024-06-18

**Authors:** Fernando Avelar-Muñoz, Roberto Gómez-Rosales, Arturo Agustín Ortiz-Hernández, Héctor Durán-Muñoz, Javier Alejandro Berumen-Torres, Jorge Alberto Vagas-Téllez, Hugo Tototzintle-Huitle, Víctor Hugo Méndez-García, José de Jesús Araiza, José Juan Ortega-Sigala

**Affiliations:** 1Unidad Académica de Física, Universidad Autónoma de Zacatecas, Campus Universitario II, Av. Preparatoria s/n, Zacatecas 98060, Zacatecas, Mexico; fernando.avelar@fisica.uaz.edu.mx (F.A.-M.); roberto.gomez@fisica.uaz.edu.mx (R.G.-R.); aaortizhernandez@uaz.edu.mx (A.A.O.-H.); javier.berumen@fisica.uaz.edu.mx (J.A.B.-T.); jvargas@fisica.uaz.edu.mx (J.A.V.-T.); tototzintle@fisica.uaz.edu.mx (H.T.-H.); araiza@fisica.uaz.edu.mx (J.d.J.A.); 2Universidad Politécnica de Zacatecas, Plan del Pardillo S/N, Parque Industrial, Fresnillo 99059, Zacacatecas, Mexico; 3Unidad Académica de Ingeniería Eléctrica, Universidad Autónoma de Zacatecas, Campus Ingeniería, Ramón López Velarde 801, Col. Centro, Zacatecas 98000, Zacacatecas, Mexico; hectorduran3@gmail.com; 4Laboratorio Nacional-CIACyT, Universidad Autónoma de San Luis Potosí, Av. Sierra Leona 550, Col. Lomas 2a. Sección, San Luis Potosí 78210, San Luis Potosí, Mexico; victor.mendez@uaslp.mx

**Keywords:** stable p-type ZnO, Ag-N doping ZnO, high hole concentration, dual doped ZnO

## Abstract

p-type Ag-N dual acceptor doped ZnO thin films with long electrical stability were deposited by DC magnetron reactive co-sputtering technique. After deposition, the films were annealed at 400 °C for one hour in a nitrogen-controlled atmosphere. The deposited films were amorphous. However, after annealing, they crystallize in the typical hexagonal wurtzite structure of ZnO. The Ag-N dual acceptors were incorporated substitutionally in the structure of zinc oxide, and achieving that; the three samples presented the p-type conductivity in the ZnO. Initial electrical properties showed a low resistivity of from 1 to 10^−3^ Ω·cm, Hall mobility of tens cm^2^/V·s, and a hole concentration from 10^17^ to 10^19^ cm^−3^. The electrical stability analysis reveals that the p-type conductivity of the ZnO:Ag,N films is very stable and does not revert to n-type, even after 36 months of aging. These results reveal the feasibility of using these films for applications in short-wavelength or transparent optoelectronic devices.

## 1. Introduction

The increasing interest that has emerged in the development of short-wavelength optoelectronic devices based on transparent semiconductors has generated the demand for the use of group II–VI semiconductors. In recent years, ZnO has been considered a promising semiconductor for the development of transparent conductive materials since it has a direct bandgap of 3.3 eV and low threshold voltage devices. Moreover, it is well known that ZnO has a large number of advantages over AlN and GaN, two commonly used semiconductors in the short-wavelength optoelectronics industry [1,2,3]. Some of these advantages include a large exciton energy (60 MeV), which can lead to violet and UV sources with high brightness and lower power thresholds at room temperature. The ZnO has greater resistance to radiation than the Si, GaAs, CdS, and GaN. Therefore, it enhances its use for space applications.

Well-known that undoped ZnO shows n-type conductivity. Some studies suggested that this conductivity is due to the presence of hydrogen in the structure, which works as a donor with a very low ionization energy (30 meV) [4,5]. This behavior is due to hydrogen tends to diffuse very easily in large quantities into the structure of ZnO, and in turn, it is present in all the techniques used for the deposition of ZnO thin films. Despite the extensive amount of work reported on p-type thin films deposited by different techniques and with different doping elements, which may enter into the substitution of oxygen or zinc, a reliable electrical conductivity for p-type ZnO has not yet been achieved [6,7,8,9,10].

Therefore, this subject is of great interest to the scientific community of materials science since the principal difficulty in developing ZnO-based optoelectronic devices lies in the manufacture of p-type thin films with good crystalline quality and acceptable electrical stability. The lack of stability is mainly due to the low solubility of the acceptor dopants, the depth of the acceptor level, and the compensation effect between the acceptor and native ZnO donor dopants [1,5,11,12]. Within all the experimental methodologies addressed for the manufacture of p-type ZnO, the double doping method, in which the dopant can be either a double acceptor (acceptor-acceptor) or acceptor-donor, has proved to be the best channel to overcome these difficulties. The double-doping method using two acceptor agents to prepare p-type ZnO, Li-N [13,14], Ag-Li [15], or Cu-N [16] dopants has been recently investigated. Additionally, experimental publications have been reported on the manufacture of p-type ZnO:Ag,N thin films with good electrical properties obtained by the ultrasonic pyrolytic spray technique [17,18,19,20]. Similarly, there are new reports using the technique of deposition by assisted ion implantation [21], the sol-gel and spin coating methods [22,23], and the sputtering method [24].

At the same time, theoretical investigations suggest that Ag and N are the two best candidates for the production of p-type ZnO, taking into account the effects of deformation and substitution energy levels of Ag-Zn and N-O [25,26,27].

In general, deposition by the sputtering technique offers several advantages over other physical vapor deposition methods, among which we can highlight high deposition speed, sample uniformity, high composition control, greater reproducibility, and superior adhesion to the substrate. In the particular case of ZnO:Ag,N thin films, the main advantage offered by the reactive DC co-sputtering technique is that this deposition process allows precise control of the composition of the deposited film since co-sputtering can deposit the different components one by one, and facilitates the incorporation of reactive gases, which are kept under control throughout the process. The deposit offers excellent uniformity, guaranteeing that the deposited film has a uniform thickness and composition throughout the substrate. Finally, the samples are highly reproducible.

This paper, there is presented a study of the electrical stability of p-type ZnO:Ag,N thin films deposited by the DC reactive magnetron co-sputtering method. After 36 months of aging, the electrical stability analysis reveals that the p-type conductivity of the ZnO:Ag,N films is very stable and does not revert to n-type. 

## 2. Materials and Methods

Ag and N double acceptor doped ZnO thin films were deposited on GaAs (100) substrates by DC magnetron co-sputtering reactive method, using metallic targets of zinc and silver as precursors, both of two inches in diameter and a purity of 99.99%. Previously to the deposit process, the growth chamber was evacuated until reaching a base pressure of 8.7 × 10^−7^ Torr. The deposits were made at a working pressure of 6 × 10^−3^ Torr, and the sputtering reactive atmosphere was composed of 5 sccm of argon (99.995%), 15 sccm of nitrogen (99.999%), and three different oxygen (99.999%) flows of 2 sccm, 2.5 sccm, and 3 sccm, respectively. All films were deposited at room temperature (approx. 25 °C) with a power density of 4 W/cm^2^ and 1.5 W/cm^2^ in zinc and silver targets, respectively. The target-substrate distance was 10 cm and 15 cm for Zn and Ag, respectively. In order to control the percentage of silver incorporated in the film during the growth process, the shutter for the silver target was opened intermittently, with a period of only one second open for every 20 s of deposit. After growth, the samples were thermally treated at 400 °C for 1 h in a nitrogen atmosphere. To determine the elemental composition of the samples, the films were characterized in a Jeol Scanning Electron Microscope, model JSM-6390LV, JEOL USA Inc., Peabody, MA, USA, which is equipped with an Energy Dispersion Spectrometer (INCA X-sight Oxford Inst. Model 7558, Oxford Instruments Analytical, High Wycombe, UK). The crystalline structure of thin films ZnO unpurified with Ag and N was studied in a D-5000 Diffractometer from Siemens AG (Bruker AXS), Munich, DEU, using the Cu K alpha radiation line with a wavelength (0.1541 nm). The Raman spectra were determined by a Raman Microprobe Spectrometer, model HORIBA LabRam HR UV 800, HORIBA France SAS, Palaiseau, FRA, equipped with a solid-state laser of 532 nm and with an Olympus BXFM-ILHS Confocal Microscope, Olympus France SAS, Rungis Cedex, FRA, with an automated XYZ platform. The photoluminescence characteristics were measured at room temperature with a fluorescence spectrometer (Edinburgh Instruments, Mod FLS980 EPLED Series, Edinburgh Instruments, Livingston, UK) equipped with pulsed LEDs at 250 nm as an excitation source with bandpass filters for spectral purity. The concentration of charge carriers was determined by the Hall Effect in an Ecopia HMS-3000 measurement system, Ecopia Corp, Incheon, Republic of Korea, using the Van der Pauw method.

## 3. Results

### 3.1. Elemental Composition

Through the results of X-ray Dispersed Energy Spectroscopy, the presence of zinc, oxygen, nitrogen, and silver was measured in all films deposited and thermally treated. The spectra obtained for each of the films are shown in Figure 1. The atomic concentrations obtained for each film are shown in Table 1. According to these values, all the samples present a slightly higher concentration of oxygen, in which a maximum value of 58.30% is observed for the films with a gas flow ratio of 5/2.5/15. On the other hand, the atomic concentration of Zn was less than 40%, with a minimum value of 36.28% for the film deposited with an oxygen flow of 3.0 sccm. The atomic percentage of nitrogen incorporated in the samples varies from 3.78 to 4.65 at. % as the flow of oxygen present in the reactive atmosphere decreases. Additionally, as the percentage of nitrogen in the sample increases, so does the percentage of silver incorporated in the different films, as can be seen in Table 1.

### 3.2. Crystalline Structure

Figure 2 shows the X-ray diffraction patterns of ZnO:Ag,N films. In all diffractograms, seven characteristic peaks of the ZnO wurtzite hexagonal structure are observed; these peaks are centered at 31.78, 34.48, 36.38, 47.59, 56.46, 62.64, and 67.72 2θ degrees, which correspond to the diffraction planes (100), (002), (101), (102), (110), (103) and (200) respectively. High intensity is observed for the (002) diffraction peak for the film deposited under a reactive atmosphere with 2.0 sccm of oxygen, indicating that the films have a preferential crystalline orientation along the c-axis perpendicular to the surface of the substrate. However, this condition is lost for the rest of the samples, which showed similar intensities for the peaks centered at 34 and 36 degrees and associated with (002) and (101) planes, respectively. As shown in the diffractograms, in the case of the samples deposited at a higher concentration of oxygen, the intensity of the (002) peak gradually decreases while the peak associated with the plane (101) increases. The leftover peaks presented in the diffraction patterns remain virtually unchanged.

### 3.3. Vibrational Properties

In Figure 3, the vibrational properties of the p-type ZnO:Ag,N thin films are presented. According to group theory, for the perfectly crystalline ZnO (wurtzite hexagonal structure belongs to the P63mc space group), the optical modes that should exist in the ZnO with wurtzite structure are given by the equation: (1)$$\Gamma_{opt}=A_{1}+E_{1}+2E_{2}+2B_{1}$$
where modes ***A_1_*** and ***E_1_*** are polar modes, with optical transverse splitting (***A_1T_*** and ***E_1T_***) and longitudinal optical splitting (***A_1L_*** and ***E_1L_***), while modes ***E_2_*** are non-polar modes and consists of two modes with low and high frequency denoted as ***E_2_^low^*** and ***E_2_^high^***, respectively. The former modes are known as active modes. Finally, two modes, ***B_1_*** and silent, are presented. Modes ***E_2_^low^***, ***E_2_^high^***, ***A_1T_***, ***A_1L_***, ***E_1T_***, and ***E_1L_*** are located at 101, 447, 381, 574, 407, and 583 cm^−1^, respectively. Now, analyzing the obtained Raman spectra, the dominant peak indicated in the Raman spectra as ***E_2_^high^*** is observed at 447 cm^−1^. This signal corresponds to the more characteristic phonon of the Wurtzite hexagonal structure of the ZnO. The peak positioned at 573 cm^−1^ is associated with the ***A_1L_*** mode, and finally, the peak positioned at 589 cm^−1^ could be associated with the E_1L_ optical mode, and it is attributed to the formation of defects such as the absence of oxygen and zinc interstitial [28,29]. The additional peak observed at 274 cm^−1^ in the Raman spectrum can be attributed to the vibration of Zn atoms, where some of its first nearest neighbor oxygen atoms are replaced by nitrogen atoms in the hexagonal structure [30]. Additional local vibrational modes (LVM) in the range from 460 to 520 cm^−1^ can be observed in the Raman patterns of ZnO:Ag,N films. These LVM are explained due to the defects induced by impurities that break the translational symmetry of the crystal [31]. A particular mode centered at 493 cm^−1^ has been reported to be related to the Ag atoms when they replaced the Zn atoms in the ZnO structure [32]. Then, after analyzing the Raman patterns for the ZnO:Ag,N thin films, the presence bands located at 274 and 493 cm^−1^ confirm that the impurities have been incorporated substitutionally into the ZnO wurtzite hexagonal structure. Finally, as can be seen in Figure 3, in the three Raman spectra, three very sharp peaks appear located at 302, 660, and 693 cm^−1^. Because of the shape of the signal, they could be considered anomalous bands and could not be associated with thin films. However, in recent studies for doped ZnO, it has been found that it is possible to observe coupled multiphonic processes such as ***TA + LO*** [33] or second-order modes associated with silent ***B_1_*** modes [34]. The band centered at 302 cm^−1^ has been assigned to ***B_1_^high^–B_1_^low^*** second-order mode. The mode at 664 cm^−1^ can be assigned to the ***TA+ B_1_^high^*** second-order mode. Furthermore, the 837 cm^−1^ mode can be attributed to the ***B_1_^low^ + B_1_^high^*** second-order mode. The reason for the observation of the second-order modes associated with silent B_1_ modes in the Raman spectra of the ZnO:Ag,N films is likely disorder-activated Raman scattering (DARS) [34,35,36,37]. This scattering is induced by the breakdown of the translation symmetry of the lattice caused by defects or impurities either because of the dopant nature or because of the growth conditions.

### 3.4. Photoluminescence Properties

Figure 4 shows the photoluminescence (PL) spectra, measured at room temperature, with pulsed LEDs of 250 nm as the excitation source. In all three PL spectra, a very strong ultraviolet emission is observed near the edge of the band (NBE). This peak of ultraviolet emission shows a direct dependence as a function of the total concentration of dopants. As is observed, this PL signal becomes wider and of lower intensity as the total percentage of dopant increases. The PL spectra of all the p-type Ag-N dual-doped ZnO thin film samples could be deconvoluted to several peaks emanating due to the transition between band edges and the defects and dopant levels [38]. Using a deconvolution of six peaks with Gaussian profiles, in the three samples it was found that this strong ultraviolet emission signal is composed of four emissions, which are free exciton emission (FX), free-electron-to-neutral acceptor emission (FA), donor-acceptor pair recombination emission (DAP) and zinc interstitial energy level (Zn_i_) to the valence band emission (Zn_i_ to VB), from lower to higher wavelength respectively [39,40]. The FX emission in the three samples is the signal emitted at the lowest wavelength (highest energy) and is located at 358.55 nm (3.45 eV), 358.71 nm (3.45 eV), 352.87 nm (3.51 eV) for the samples deposited from the lowest to highest oxygen flux, respectively. The FA emission is located at 384.95 nm (3.22 eV), 380.06 nm (3.26 eV), and 387.62 nm (3.20 eV) for the samples deposited from the lowest to highest oxygen flux, respectively. However, this signal is one of those that show a greater dependence on the percentage of dopant. As the total percentage of dopant in the sample increases, the intensity and width of the peak increase. For the film with a lower dopant concentration, the DAP emission is centered at 391.01 nm (3.17 eV); however, the edge of this emission band shifts towards higher wavelengths (lower energies) as the dopant atomic percentage increases. For the other two samples, the UV emission peaks are centered at 393.58 nm (3.15 eV) and 398.82 nm (3.12 eV), respectively. Finally, the observed violet emission, due to transition energy from zinc interstitial energy level (Zn_i_) to the valence band, at 411.29 nm (3.01 eV), 419.89 nm (2.95 eV) and 435.55 nm (2.84 eV) for the samples deposited from the lowest to highest oxygen flux, respectively; furthermore, this peak also shows a high dependence on the percentage of total dopant incorporated in the samples, since as the percentage of total dopant in the sample increases, the intensity of this signal decreases drastically. Recombination between the zinc interstitial energy level (Zn_i_) and the zinc vacancy energy level (V_Zn_) contributes to blue emission at 482.75 nm (2.57 eV) and 474.02 nm (2.62 eV). This emission can only be visible for the sample with a lower concentration of dopants. Intrinsic donor defects such as oxygen vacancies (V_O_), which act as donor defects, contribute to 694.58 nm emission [41]; however, in the three samples studied, it can be observed that these last two mentioned emissions are very weak compared to the emission in the ultraviolet, that is, there is a low density of native defects, which is in good agreement with the results of XRD and Raman spectra.

### 3.5. Electrical Properties

The characterization of the ZnO:Ag,N thin films by Hall Effect in Van der Pawn configuration indicates that all the studied samples exhibit a p-type conductivity, with relatively high densities of charge carriers, as shown in Table 2. According to the values obtained, it can be seen that the film deposited at a lower concentration of oxygen presents a lower carrier density. Moreover, the concentration of charge carriers increases as the oxygen ratio in the sputtering atmosphere increases. This behavior can be explained if it is considered that in O-poor, the incorporation of nitrogen and silver is favored, and with this, the acceptor levels are generated with the substitution of zinc and oxygen by silver and nitrogen, respectively. 

In order to verify the stability in time of the p-type conductivity of the ZnO:Ag,N thin films, the electrical measurements were carried out once the samples were deposited and after 12 and 36 months of aging. During this time, the samples were stored at atmospheric pressure in ordinary individual plastic containers. 

As seen in Figure 5, after 36 months, all samples still sustain p-type conductivity. For the film deposited at a higher oxygen concentration, a slight decay of almost an order of magnitude in the free carrier density is observed. On the other hand, for the other two studied samples, practically the same values for the hole concentration were obtained after the 36 months had elapsed. This result shows that ZnO:Ag,N films had a minimal deterioration in their electrical properties, that is to say, that the free carrier density is practically constant, in particular for films deposited at a flow of 5 sccm of Ar, 3 sccm of O_2_ and 15 sccm of N_2_. Finally, the results ensure that the p-type conductivity of ZnO:Ag,N films is very stable and does not revert to type n, even after 36 months, as is usually the case with simple doped ZnO:Ag or ZnO:N. Comparing this result with previous reports, p-type conductivity was frequently reported with stability of one year [23,42]; there is only one work focused on the N-B co-doped ZnO film that reports stability in conductivity of two years [43], which indicates that this issue still remains an open problem since greater stabilities in p-type conductivity are required for an optoelectronic device to have affordable durability. With the increase in the stability of ZnO p-type thin films, progress is being made in the possibility of generating longer-lasting optoelectronic devices based on zinc oxide, which has been one of the most sought-after objectives in the last 20 years within the II–VI semiconductor device research community. According to these results, we can thus affirm that the deposited films maintain the electrical properties almost without evident degradation, which reveals that the self-compensation introduced by intrinsic defects in the films is suppressed due to the double acceptor doping.

### 3.6. Band Structure and DOS

Using the wurtzite ZnO unit cell as the elemental basis, the unit cell (4 atoms) and 5 supercells (with 32 atoms each) were used as ZnO pure and doped. This method is implemented and fully integrated in the QUANTUM ESPRESSO suite of codes [44,45] (https://www.quantum-espresso.org/, accessed on 15 April 2024) for plane wave and norm-conserving pseudopotentials [46,47]. The exchange-correlation potential was determined using the generalized gradient approximation considering the Perdew-Burke-Ernzerhof scheme (GGA-PBE). The electron wave function was expanded in plane waves with a cut-off Energy of 80 Ry (1088.5 eV) and 560 Ry (4353.8 eV) for the charge. The used Monkhorst-Pack grid was 6 × 6 × 4 for the irreducible Brillouin zone sampling. Self-consistency in total energy was achieved with a tolerance of less than 10^−6^ Ry (1.36 × 10^−5^ eV).

For the analysis, firstly, it can be shown how the density of states is affected by the presence of the impurities, Figure 6. As the impurities are integrated neatly to the top of the valence band, in the band gap, there are states associated with nitrogen, silver, and the hybridization of both (N and Ag) with zinc and oxygen. The major effect is when both impurities are integrated into the structure. It can be shown that the principal value (maximum) is located below the mid-half of the band gap, which means they are type-p doping impurities. This result fits perfectly with the shift and broadening observed for the strong ultraviolet emission (NBE) of the photoluminescence spectra presented in Figure 4, and it is evidence of why the structure has enough p-type carriers to remain stable and not reverse, as has been reported in other cases. 

The effect of the impurities in the band structure is evident in Figure 6, where there are 4 zone bands: the first is associated with the O2s, around the −17 to the −15 eV. The second, very subtle, due to the N2s contributions, around the −13 to the −12 eV, the principal associated to the Zn3d, O2p, and Ag 4d, ending in the 0eV for ZnO, and the effects due to hybridization among Zn, O, N and Ag with their hybridizations. The conductance is the upper band, and it is due principally to Ag5d, Zn4d, N2p, and O2p. The bands are forced, as impurities are integrated, to have higher energy values. The edge of the density of states for the case in which it is doped with one silver and one nitrogen atom is shifted 0.376 eV regarding the edge of undoped ZnO, while when it is doped with one silver and three nitrogen atoms the shift is up to 0.745 eV. It can be shown in Figure 7.

## 4. Conclusions

ZnO:Ag,N films annealed at 400 °C are crystalline with a wurtzite hexagonal structure typical for ZnO. As a result of the incorporation of the double acceptor agent, the films have p-type conductivity. Silver and nitrogen enter the structure as substitutes for zinc and oxygen, respectively, forming Ag-Zn and N-O acceptor sites, whereby a double contribution is generated for the creation of holes, thereby achieving a high density of holes. The electrical stability analysis reveals that the p-type conductivity of ZnO:Ag,N thin films is very stable, and it does not revert to n-type, even after 36 months, as is usual in the case with simple doped ZnO:Ag or ZnO:N. According to these results, it is possible to affirm that ZnO:Ag,N films, deposited and annealed at 400 °C, maintain the electrical properties almost without evident degradation, which reveals that the self-compensation introduced by intrinsic defects in the films is suppressed due to double acceptor doping.

## Figures and Tables

**Figure 1 micromachines-15-00800-f001:**
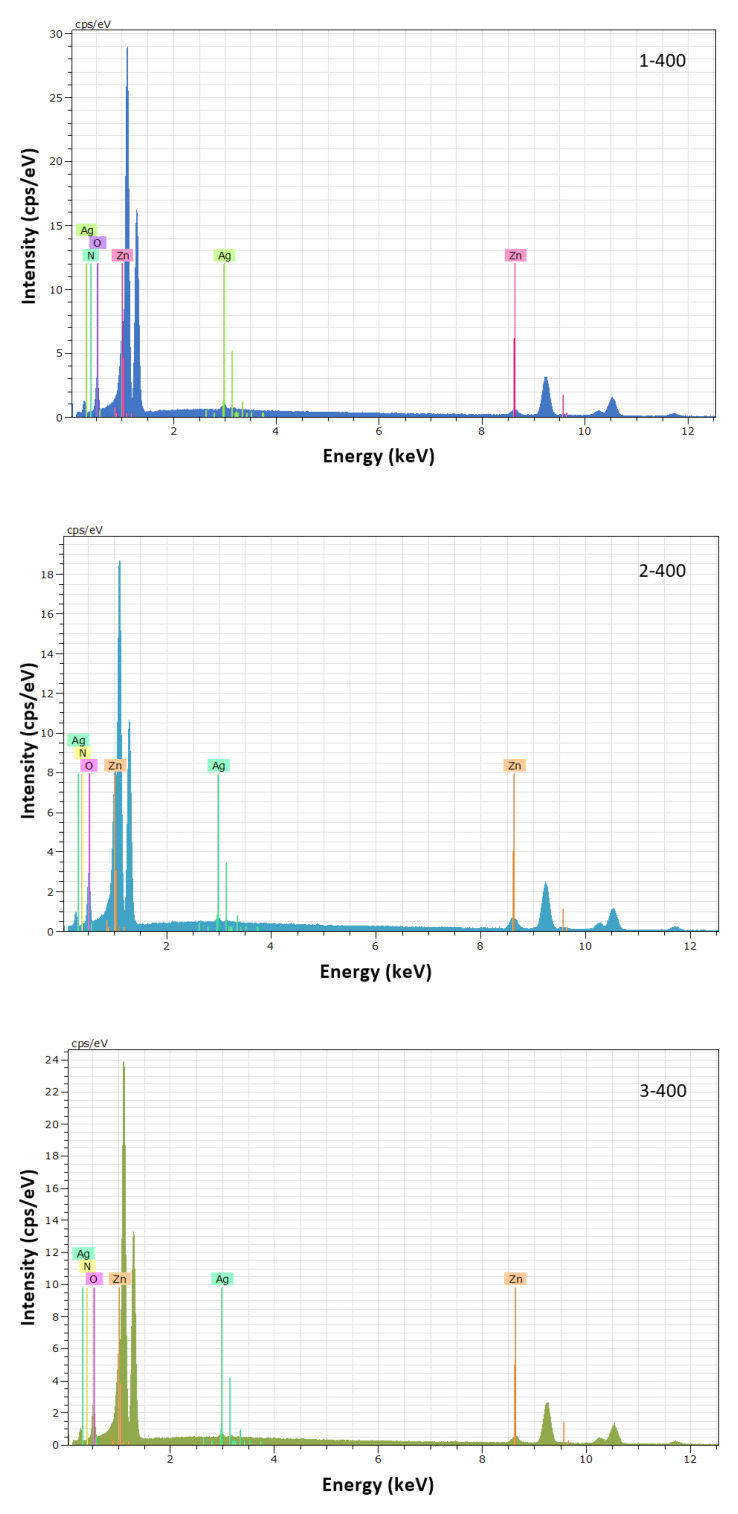
X-ray dispersed energy spectrums of p-type ZnO:Ag,N thin films deposited under a sputtering reactive atmosphere of 5 Ar/3.0 O_2_/15 N_2_ (1-400), 5 Ar/2.5 O_2_/15 N_2_ (2-400) and 5 Ar/2.0 O_2_/15 N_2_ (3-400).

**Figure 2 micromachines-15-00800-f002:**
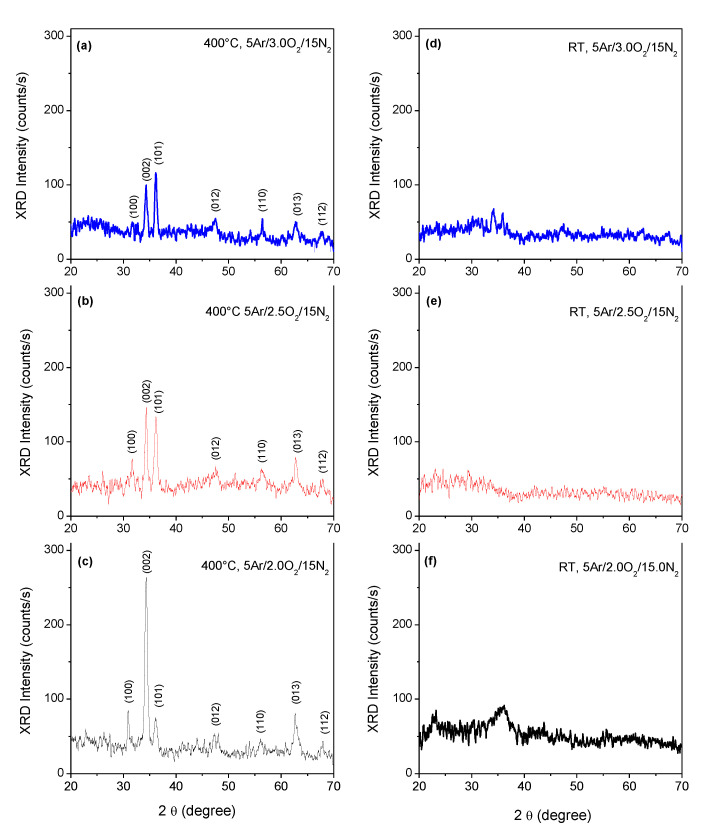
X-ray diffraction patterns of p-type ZnO:Ag,N thin films deposited under a sputtering reactive atmosphere of (**a**) 5 Ar/3 O_2_/15 N_2_, (**b**) 5 Ar/2.5 O_2_/15N_2_ and (**c**) 5 Ar/2 O_2_/15 N_2_ and annealed after deposition at 400 °C, (**d**) 5 Ar/3 O_2_/15 N_2_, (**e**) 5 Ar/2.5 O_2_/15N_2_ and (**f**) 5 Ar/2 O_2_/15 N_2_ as deposited without annealing treatment.

**Figure 3 micromachines-15-00800-f003:**
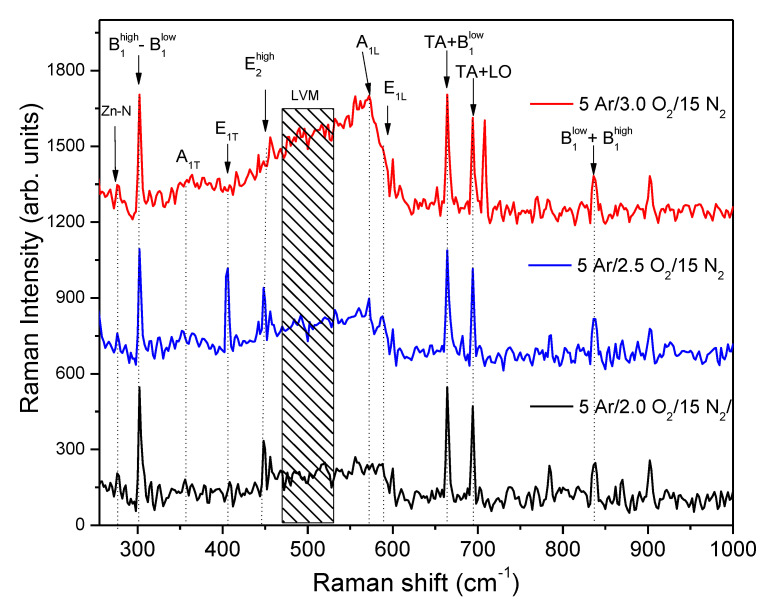
Raman spectra of p-type ZnO:Ag,N thin films deposited under a sputtering reactive atmosphere of 5 Ar/3 O_2_/15 N_2_, 5 Ar/2.5 O_2_/15 N_2,_ and 5 Ar/2 O_2_/15 N_2_ and annealed after deposition at 400 °C.

**Figure 4 micromachines-15-00800-f004:**
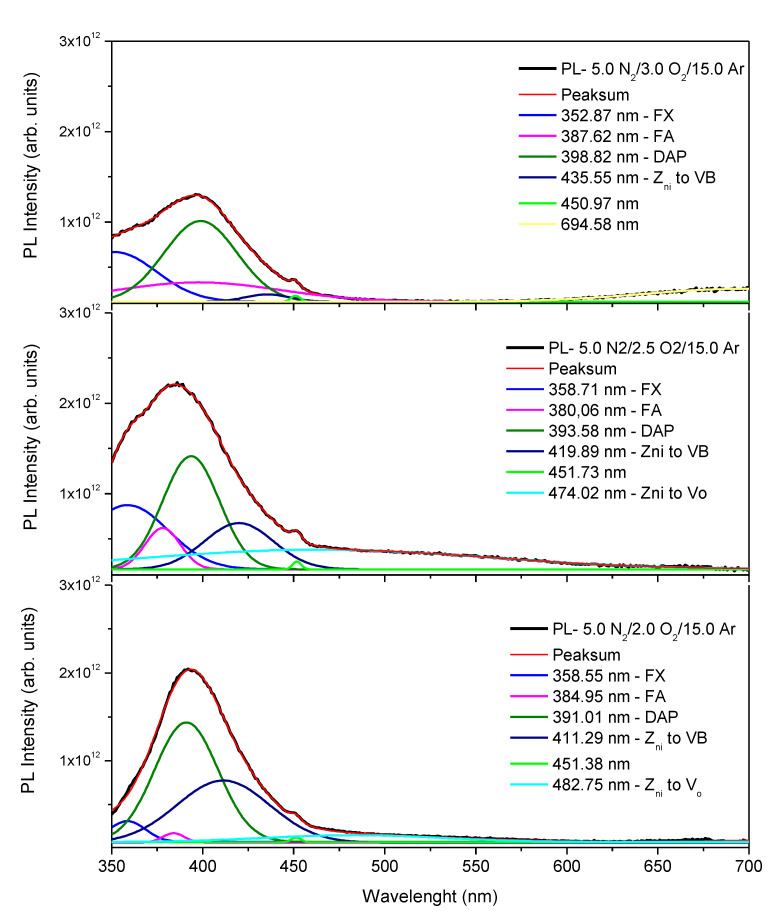
Photoluminescence spectra of p-type ZnO:Ag,N thin films deposited under a sputtering reactive atmosphere of 5 Ar/3 O_2_/15 N_2_, 5 Ar/2.5 O_2_/15 N_2,_ and 5 Ar/2 O_2_/15 N_2_ and annealed after deposition at 400 °C.

**Figure 5 micromachines-15-00800-f005:**
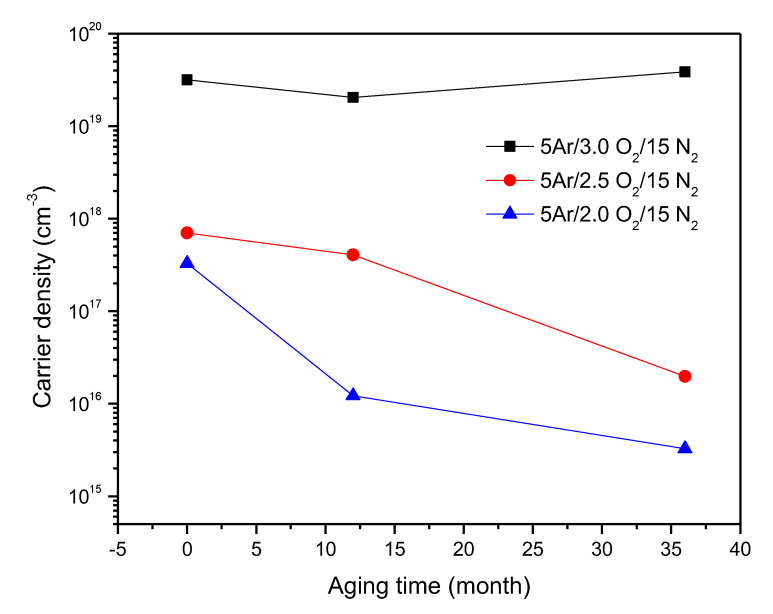
The carrier concentration of the p-type ZnO:Ag,N thin films as a function of aging time, measurements were made when it was deposited (0 months), at twelve months, and after 36 months elapsed.

**Figure 6 micromachines-15-00800-f006:**
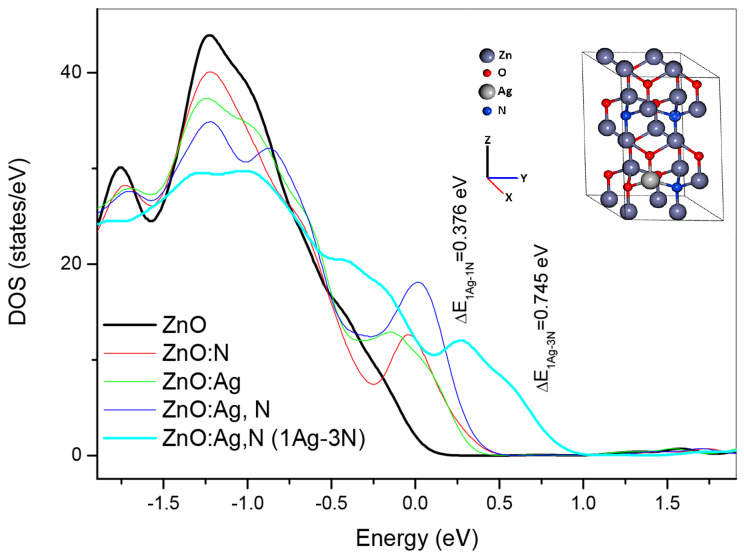
Effect of the impurities on the Density of states for the different single and dual doped p-type zinc oxide compounds.

**Figure 7 micromachines-15-00800-f007:**
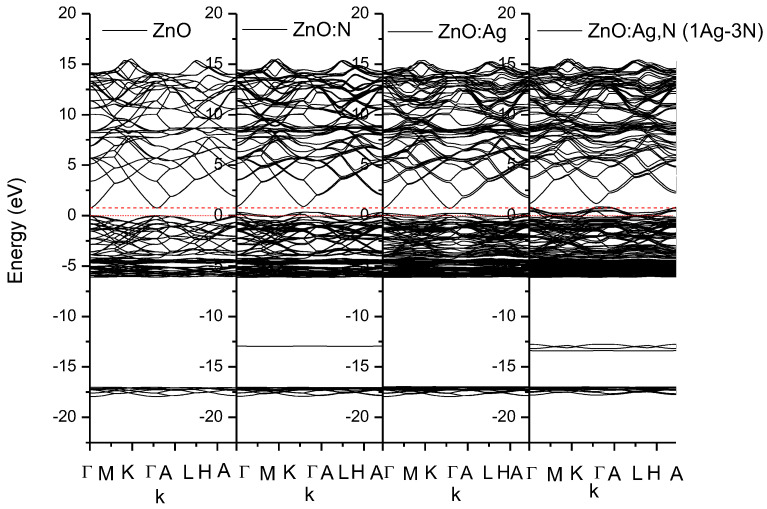
Effect of the impurities on the formation of energy bands for the different single and dual doped p-type zinc oxide compounds.

**Table 1 micromachines-15-00800-t001:** Elemental atomic concentration for the p-type ZnO:Ag,N thin films deposited under three different sputtering reactive atmospheres and annealed at 400 °C.

Film	Gas Flow Ratio Ar/O_2_/N_2_ (sccm)	Atomic Percentage (%)			
		Zn	O	N	Ag
1-400	5.0/3.0/15.0	36.28	57.54	4.65	1.56
2-400	5.0/2.5/15.0	37.27	58.30	3.78	0.59
3-400	5.0/2.0/15.0	39.56	55.60	3.80	1.04

**Table 2 micromachines-15-00800-t002:** Initial values of the carrier density, mobility, and resistivity for the p-type ZnO:Ag,N thin films.

Film	Gas Flow Ratio Ar/O_2_/N_2_ (sccm)	Initial Values			
		Carrier Density (cm^−3^)	Mobility (cm^2^/V.s)	Resistivity (Ω·cm)	Type Conductivity
1-400	5.0/3.0/15.0	3.290 × 10^17^	16.05	1.182 × 10^−1^	p-type
2-400	5.0/2.5/15.0	7.008 × 10^17^	20.46	4.33 × 10^−1^	p-type
3-400	5.0/2.0/15.0	3.174 × 10^19^	22.99	8.555 × 10^−3^	p-type

## Data Availability

All data are available on request from the authors.

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
