# Peer review of "Long Electrical Stability on Dual Acceptor p-Type ZnO:Ag,N Thin Films"

_micromachines, 2024, doi:10.3390/mi15060800_

Round 1
Reviewer 1 Report
Comments and Suggestions for Authors
This study was well-written with an interesting idea that the Ag-N doped ZnO could be a stable p-type semiconductor for future works. However, several experiments should be done to improve the scientific content before accepting for publication.
1. Author should add XRD pattern of sputtered film without annealing for the statement "The deposited films were amorphous. However, after annealing they crystallize in the typical hexagonal wurtzite structure of ZnO". The reference doi.org/10.3390/nano13222979, doi.org/10.1016/j.rinp.2019.102159 may be helpful.
2. In Fig. 3, the PL spectra require a fitting analysis for determining energy-state levels of defects in prepared Ag-N doped ZnO film, like references doi.org/10.1007/s00339-020-03767-0, doi.org/10.1021/acsomega.3c05266, 10.1039/c9ra06480j, doi.org/10.1021/acsaelm.3c01332.
3. The trap densities in each film should be quantitative by SCLC measurement like previous research such as 10.1039/C9TA14207J, doi.org/10.1021/acsenergylett.9b02720, 10.1039/D4TA00333K.
4. Theoretically, an effective p-type film can couple with a n-type film to achieve a good p-n contact in practice. So, author should add a current-voltage (I-V) curve of a p-n structure (e.g. their p-type ZnO/undoped n-type ZnO) in dark condition. If the curves show an asymmetrical shape, it will be a solid evidence for the p-type nature of prepared Ag-N doped ZnO. Please refer to other publications such as 10.1016/j.phpro.2014.07.010, 10.1016/j.mssp.2010.05.005, doi.org/10.3390/coatings13050921, 10.1002/admt.201700208, 10.1063/1.3078806, doi.org/10.1016/j.jallcom.2023.172422
Author Response
We thank you very much for the time dedicated to this review; we truly consider the work you do as a reviewer invaluable. After analyzing each observation made in this document, we addressed item Q1 and Q2 to the best of our ability and capacity; however, items Q3 and Q4 were not addressed since these films were deposited without bottom electric contact; however, along with this study, we have been developing a second work focused on the growth and characterization of a p-n homojunction using precisely the conditions of these samples for the preparation of the p-type film of the structures, which we hope
to publish in the coming months.
The point by point response is presented in the attached document

Reviewer 2 Report
Comments and Suggestions for Authors
F. Avelar-Muñoz et al presents an article entitled “Long electrical stability on dual acceptor p-type ZnO:Ag,N thin films” where authors report a dual acceptor p-type ZnO:Ag,N thin films, demonstrating electrical properties, achieving low resistivity of from 1 to10−3 Ω.cm, Hall mobility of tens cm2/V.s and a hole concentration from 1017 to 1019 cm−3. Based on the analysis of the manuscript, the following modifications are recommended to enhance its quality and impact:
1. Despite the existence of simpler synthesis techniques for ZnO:Ag, N, why did the author choose the DC reactive magnetron co-sputtering technique for thin film deposition? In other words, what specific advantages does DC reactive magnetron co-sputtering offer over other simpler synthesis methods for this particular application?
2. How do the stability and electrical characteristics of these films improve the feasibility and performance of devices compared to previous materials? Please comment on it.
3. It is suggested to include the specifications for Raman and PL spectroscopy, as they are currently missing from the materials and methods section. Please include them.
4. On page 3, line 97, please include X-ray dispersed Energy spectra plots for all three samples.
5. It is suggested to include subscripts in the appropriate places in the caption of Figure 1. For example, 'O2' and 'N2' should be replaced by 'O₂' and 'N₂', respectively.
6. In Figure 2, some Raman peaks are not labeled and explained, such as the peak at approximately 410 cm⁻¹ for sample 5Ar/15N₂/2.5O₂ and peaks in the range of 650-950 cm⁻¹. Additionally, please replace commas with decimal points when labeling the samples on the plot, i.e., label the sample as 5Ar/15N₂/2.5O₂ instead of 5Ar/15N₂/2,5O₂.
7. Please mention the excitation wavelength used for PL spectra measurements shown in Figure 3.
8. Figures 3 and 4 lack major and minor tick labels. Please consider replotting the graphs with appropriate tick labels for clarity.
9. On page 6, line 173, It is suggested to modify the sentence “___with the results of XRD and Raman results” to ___with the results of XRD and Raman spectra”
Comments on the Quality of English LanguageThe proficiency in English is adequate, though there are areas that could be refined.
Author Response
We thank you very much for the time dedicated to this review; we truly consider the work you do as a reviewer invaluable. After analyzing each observation made in this document, we addressed all items to the best of our ability and capacity.
The point-by-point answer is included in the attached file.

Round 2
Reviewer 1 Report
Comments and Suggestions for Authors
One issue related to Q2 should be addressed before acceptance
In Q2, the PL (Figure. 4) did not show energy-state level related to Ag:N while the DOS (Figure. 6) stated that "the principal value (maximum) is located below the mid-half of the band gap, it means, they are shallow type p-doping impurities". Why is there the difference between the experimental data and simulated data?
Additionally, the cumulative fit should be added to Figure. 4 to demonstrate the quality of deconvoluting analysis (refer to doi.org/10.1007/s10854-021-07461-6).
Comments on the Quality of English Language
Some typos need to be revised. For example, "Zni", "VZn"...
Author Response
Thank you very much for the comment and suggestion, we addressed all observations to the best of our ability and capacity
- Comment: One issue related to Q2 should be addressed before acceptance, the PL (Figure. 4) did not show energy-state level related to Ag:N while the DOS (Figure. 6) stated that "the principal value (maximum) is located below the mid-half of the band gap, it means, they are shallow type p-doping impurities". Why is there the difference between the experimental data and simulated data?, Additionally, the cumulative fit should be added to Figure. 4 to demonstrate the quality of deconvoluting analysis (refer to doi.org/10.1007/s10854-021-07461-6).
Response: In response to the recommendation, Figure 4 was modified, in each subgraph six peaks with Gaussian profiles were identified, and the cumulative fit was also added to Figure 4. Additionally, the analysis text of Section 3.4 has been modified

Round 3
Reviewer 1 Report
Comments and Suggestions for Authors
I am pleased to accept this manuscript for publication.